# Super-Toughened Poly(lactic Acid) with Poly(ε-caprolactone) and Ethylene-Methyl Acrylate-Glycidyl Methacrylate by Reactive Melt Blending

**DOI:** 10.3390/polym11050771

**Published:** 2019-05-01

**Authors:** Ao-Lin Hou, Jin-Ping Qu

**Affiliations:** National Engineering Research Center of Novel Equipment for Polymer Processing, Key Laboratory of Polymer Processing Engineering, Ministry of Education, Guangdong Provincial Key Laboratory of Technique and Equipment for Macromolecular Advanced Manufacturing, School of Mechanical and Automotive Engineering, South China University of Technology, Guangzhou 510640, China; houaolin2016@163.com

**Keywords:** poly(lactic acid), reactive melt blending, microstructure, compatibilization, toughness

## Abstract

In recent years, poly(lactic acid) (PLA) has attracted more and more attention as one of the most promising biobased and biodegradable polymers. However, the inherent brittleness significantly limits its wide application. Here, ternary blends of PLA, poly(ε-caprolactone) (PCL) with various amounts of ethylene-methyl acrylate-glycidyl methacrylate (EMA-GMA) terpolymer were fabricated through reactive melt blending in order to improve the toughness of PLA. The effect of different addition amounts of EMA-GMA on the mechanical properties, interfacial compatibility and phase morphology of PLA/PCL blends were studied. The reactions between the epoxy groups of EMA-GMA and carboxyl and hydroxyl end groups of PLA and PCL were investigated thorough a Fourier transform infrared (FT-IR). The miscibility and thermal behavior of the blends were studied through a dynamic mechanical analysis (DMA), differential scanning calorimetric (DSC) and X-ray diffraction (XRD). The phase morphology and impact fracture surface of the blends were also investigated through a scanning electron microscope (SEM). With the addition of 8 phr EMA-GMA, a PLA/PCL (90 wt %:10 wt %)/EMA-GMA ternary blend presenting a suitable multiple stacked phase structure with an optimum interfacial adhesion exhibited an elongation at break of 500.94% and a notched impact strength of 64.31 kJ/m^2^ with a partial break impact behavior. Finally, the toughening mechanism of the supertough PLA based polymers have been established based on the above analysis.

## 1. Introduction

Due to the constant consumption of traditional plastic, diminishing oil reserves and increasing plastic waste will threaten the ecological environment. In recent years, considerable attention has been paid to biodegradable polymers, mainly owing to an increasing interest in the preservation of the environment and the substitution of petrochemical polymers [1,2].

Poly(lactic acid) (PLA), an aliphatic polyester, one of the most promising and potential renewable bio-based polymers produced from renewable sources such as corn [3], has been widely applied in biomedical applications, such as the drug delivery system and surgical suture [4], because of its excellent biocompatibility, good biodegradability and high mechanical strength [5,6]. However, the inherent deficiencies of PLA, including its natural brittleness and poor elongation at break, narrow processing window and low melt strength, largely limit its further application [7]. The brittleness in particular restricts its use in many fields. Thus, there is an urgent need to improve the brittleness of PLA. A wide and effective method to tailor the material’s properties, while making use of what is commercially available, is by melt blending. Until now, binary blends of PLA with other ductile polymers have been widely researched [8,9,10,11,12,13,14]. However, most blends suffer from a lower toughness due to the poor miscibility between the two immiscible phases. A practical and useful way to improve the toughness and flexibility of PLA is multicomponent polymer blending. The blends, which consist of at least three components, integrate the advantages of each component to enhance the physical properties [15,16,17]. Zhang et al. considered a ternary blend system consisting of PLA, an epoxy-containing elastomer (EBA-GMA), and a zinc ionomer (EMAA-Zn) to improve the toughness of PLA [18]. The zinc ions promoted the reactive compatibilization of PLA and the elastomer, and a “salami”-like phase structure resulting in the excellent impact strength of PLA. Ravati et al. investigated the quantitative interfacial coarsening for a series of ternary blends [19,20]. They found that changing the components and component ratio could help control the morphologies of the blends. Therefore, multicomponent polymer blending is a preferred method of modifying PLA in order to obtain excellent properties.

With a view to protect the environment against pollution, biodegradability is without doubt the most valuable feature. At this point, environment-friendly biodegradable polymers will be ideal as modifiers for PLA. Blending with a ductile polymer, poly(ε-caprolactone) (PCL), another aliphatic polyester, which is also biodegradable and biocompatible, is one of the most promising choices [21,22]. The interfacial reactive compatibility of PLA blends and the formation of special phase morphologies play key roles in achieving the high impact strength of PLA blends. However, PLA and PCL are thermodynamically incompatible with each other; PLA blends with PCL have been reported as being phase-separate systems with poor interfacial adhesion [23,24]. Many studies have been done on enhancing the compatibility of the PLA/PCL blend by using the addition of reactive compatibilization methods. Different reactive compatibilizers have been used improve the compatibility PLA/PCL blend. Semba et al. added dicumyl peroxide (DCP) to PLA/PCL to improve the blend component compatibility by inducing a chemical interfacial cross-linking; the impact strength of the optimum composition was 2.5 times superior to neat PLA, and a plastic deformation was observed at its fracture surface [25]. Shin et al. selected glycidyl methacrylate (GMA) as a compatibilizer, and the morphological study clearly showed a reduced particle size of dispersed PCL domains and a significantly improved interfacial adhesion by the electron-beam irradiation with the addition of the GMA [26]. Another reactive compatibilizer that was applied was lysine triisocyanate (LTI). Takayama et al. used LTI as a compatibilizer in PLA/PCL blends, and they found that the fracture toughness was effectively improved [27]. Harada et al. used LTI to improve the compatibility of PLA/PCL, and the impact strength increased considerably at 20 wt % PCL and 0.5 phr LTI, while the tensile strain increased by 270% [28]. These limited improvements in toughness are not enough for the desired high toughness requirements.

In the present work, we fabricated super-tough PLA blends through multicomponent polymer reactive melt blending of bio-based PLA with biodegradable PCL and a functionalized copolymer ethylene methyl acrylate-glycidyl methacrylate (EMA-GMA) elastomer that contains epoxy groups. As far as we know, such a novel PLA ternary blend was not reported before. The mechanical performance, interfacial compatibilization and phase morphology of multicomponent blends were thoroughly investigated in this work.

## 2. Materials and Methods 

### 2.1. Materials

The PLA used in the study was commercial grade (PLA 4032D) and obtained from NatureWorks LLC (Blair, NE, USA). Its glass transition temperature was T_g_ ≈ 60 °C, and its melting temperature was T_m_ ≈ 170 °C.

The CapaTM 6500 poly(ε-caprolactone) (PCL) was purchased from The Perstorp Group in Sweden. This PCL grade is characterized by a melt flow index of 5.60–7.90 g (10 min) at 160 °C, a density of 1.1 g/cm^3^ and a molecular weight of 50,000 g/mol. Its melt point is located in the range of 58–60 °C, and it offers a high elongation at break of more than 800%. 

Ethylene-methyl acrylate-glycidyl methacrylate (EMA-GMA) terpolymer containing 8% of glycidyl methacrylate is a product of the Arkema Company with the trade name Lotader AX 8900, obtained from Arkema Investment Co., Ltd. (Shanghai, China) 

### 2.2. Blend Preparation

PLA and PCL pellets were previously dried under a vacuum at 80 and 45 °C, respectively, for 12 h before the melt blending. The blending was carried out using a Brabender Plasticorder with a chamber volume of 55 cm^3^ and roller blades at 200 °C for 9 min with a rotation speed of 60 rpm. The blend ratio was controlled at 90:10 wt %, and EMA-GMA was added at 0, 2, 4, 6, 8, and 10 phr. Furthermore, the pure PLA was subjected to the same mixing treatment to create the same thermal history as that of the blends. The abbreviation for each formulation is assigned as follows: PLA, PLA/PCL, PLA/PCL/2, PLA/PCL/4, PLA/PCL/6, PLA/PCL/8, PLA/PCL/10, respectively. Then, all of the samples were hot-compression-molded into 1- or 4-mm-thick sheets at 200 °C and 12 MPa for 6 min. All the samples were conditioned at 50% relative humidity and 25 °C for at least 48 h before the testing.

### 2.3. Fourier Transform Infrared (FT-IR) Spectroscopy 

The FT-IR spectroscopy (Nicolet Nexus 670, Dongguan, China) was used to study the molecular group reactions between PLA and PCL with EMA-GMA. Take a small amount of PLA, PCL, EMA-GMA, PLA/PCL, PLA/PCL/EMA-GMA samples, and press the samples into films with a hot stamper. Scan each film with FT-IR spectroscopy with 64 scans at a range of 400 to 4000 cm^−1^.

### 2.4. Dynamic Mechanical Analysis (DMA)

DMA was conducted on a Netzsch DMA242c instrument (Selb, Germany) using the method of three-point bend. The dynamic loss (tanδ) and the storage modulus were determined at a frequency of 1 Hz and a heating rate of 3 °C /min as a function of the temperature from −110 to 120 °C.

### 2.5. Differential Scanning Calorimetry (DSC)

DSC was performed using a Netzsch DSC instrument (model 204c, Selb, Germany) equipped with a liquid nitrogen-cooling accessory. The sample was first heated from room temperature to 200 °C at a rate of 20 °C /min, followed by an isothermal step for 3 min to eliminate previous thermal history, and then cooled to −40 °C at a rate of 10 °C/min. The second heating scans were monitored between −40 to 200 °C at a heating rate of 10 °C /min to determine the cold crystallization temperature (T_cc_), crystallization temperature (T_c_), and melting temperature (T_m_).

### 2.6. X-Ray Diffraction (XRD)

A D8 ADVANCE instrument (Bruker, Karlsruhe, Germany) was used for the X-ray diffraction (XRD) analysis on all blends. The scans were made between Bragg angles of 10° and 50° at a scanning rate of 3°/min.

### 2.7. Mechanical Testing 

The tensile measurements were carried out using an Instron universal machine (model 5566, Norwood, MA, USA) with a crosshead speed of 20 mm/min according to GB/T1040-2006. Notched Izod impact strength tests were performed on the PIT501B-Z pendulum impact tester (Zwick5117, Zwick GmbH, Ulmer, Germany) according to GB/T 1483-1996. More than five samples were examined for each formulation.

### 2.8. Scanning Electron Microscopy (SEM)

The morphology of the cryo-fractured surfaces, cryo-fractured surfaces after being etched and impact surfaces of the blends were observed by SEM imaging (S-3700N, Hitachi, Tokyo, Japan). Each sample was fractured after being immersed in illiquid nitrogen for 40 min. After this, all of the surfaces were coated with a 15 nm thick gold layer to provide enhanced conductivity, and subjected to SEM imaging at an applied voltage of 5 kV and a working distance of ~11 mm.

## 3. Results and Discussion

### 3.1. Reactive Interfacial Compatibilization

During the reactive melt blending, epoxy groups of the EMA-GMA could react with the carboxyl and hydroxyl end groups of PLA and PCL, as previous literatures discussed [29,30,31]. The FT-IR analysis was employed to prove that the reaction existed in the PLA/PCL/EMA-GMA blend. Figure 1 displays the FT-IR spectra of the PLA, PCL, EMA-GMA, PLA/PCL and PLA/PCL/8 blends. For the PLA, the characteristic peak of the hydroxyl group (-OH) was observed at 3654 cm^−1^ and 3504 cm^−1^, and for the PCL, it were observed at 3440–3630 cm^−1^ Two distinct peaks at 2995 cm^−1^ and 2945 cm^−1^ were assigned to the asymmetrical and symmetrical -CH3 stretching of PLA, while those of PCL were at 2945 cm^−1^ and 2870 cm^−1^. In all of the spectra, the peaks at 1730–1765 cm^−1^ were attributed to the carboxylic stretching, while the peaks at 1080–1250 cm^−1^ represented the asymmetric C-O-C stretching. In addition, the peaks at 1360–1460 cm^−1^ represented the -CH deformation, and those at 732–756 cm^−1^ represented the skeletal vibration of -CH2. A characteristic peak of the epoxy group at 911 cm^−1^ appeared in the spectra of EMA-GMA. Compared to the spectra of the PLA/PCL blend, the absorbance peak of -OH almost disappeared in the spectra of PLA/PCL/8, and new absorption peaks that appeared at 2852 cm^−1^, 2922 cm^−1^ in the spectra of PLA/PCL/8 were assigned to the symmetrical -CH2 stretching and asymmetrical -CH2 stretching of EMA-GMA. However, the characteristic peak of the epoxy group at 911 cm^−1^ in EMA-GMA was not found in the spectra of PLA/PCL/8. The above analysis indicated that the reaction between the carboxyl and hydroxyl end groups of the PLA, PCL and epoxy groups of EMA-GMA occurred at a high temperature.

Figure 2 shows the trends of the melting torque evolution as a function of the processing time for the PLA/PCL blends with different additions of EMA-GMA. At the beginning of the evolution, there were sharp peaks in the torque curves owing to the melting of the solid pellets. The peak value decreased along with the increasing EMA-GMA addition due to the elasticity of EMA-GMA at room temperature. Unlike the torque value of the PLA/PCL blend being relatively low and decreasing gradually, it could be observed in the magnification of the curves in the range of 6–12 min that the equilibrium torque value increased with the increasing EMA-GMA loading. In particular, for the PLA/PCL/8 and PLA/PCL/10 blends, torque rose markedly in 7–9 min. These results were attributed to the reaction between the EMA-GMA with PLA and PCL, which resulted in the grafted copolymers or cross-linked structures causing the increase of torque. Torque is strictly related to viscosity which is associated to their molecular weights [32]. In some way, the increasing equilibrium torque value reflected the higher molecular weights, and this in turn supported the FT-IR analysis that reactions occurred during the processing [33].

The possible reactions of the PLA/PCL blends with the introduction of EMA-GMA are illustrated in Figure 3. As discussed in the FT-IR analysis, the melt blending at a high temperature promoted the reactions of EMA-GMA with the PLA and PCL phase of the blend. It is believed that the reaction resulted in the grafting of EMA-GMA onto the PLA and PCL, forming PLA/PCL-graft-EMA co-polymers at the PLA-PCL interface. Additionally, the co-polymers located at the interface served as bridges connecting the PCL and PLA phase; in addition, the number of co-polymers increased with a higher EMA-GMA addition. Therefore, the interfacial adhesion between the two immiscible phases may be remarkably improved with the increasing EMA-GMA addition.

The miscibility of different components in the PLA/PCL/EMA-GMA blends has a great influence on the physical properties of the composite, so DMA was employed to investigate the molecular interaction of the different components. Table 1 summarizes the glass transition temperature (T_g_) which corresponds to the peak temperature of the curves. Figure 4 shows the Tan δ curves of neat PLA and the blends. The PLA/PCL exhibited two peaks at −46.8 °C and 80.8 °C, attributed to the glass transition of PCL and PLA, respectively. The T_g_ (80.8 °C) of PLA in the PLA/PCL blend was slightly lower than the T_g_ (81.4 °C) of the neat PLA, indicating that there was some weak molecular interaction between the two components due to the poor miscibility. As for the ternary blends, it was observed that the two peaks of each sample shifted increasingly toward each other with the increase of the EMA-GMA. The lower T_g_ of the PCL phase in the blends changed markedly from −46.8 °C to −32 °C, while the higher T_g_ of the PCL phase in the ternary blends decreased slightly compared with the PLA/PCL blend. The T_g_ change tendencies of the ternary blends indicated an improved miscibility, which resulted from the interfacial interaction between the epoxy group of EMA-GMA with the end group of the PCL and PLA, confirmed through the FT-IR results. In addition, an obvious difference between the peak intensity of the PCL phase in the blends was noticed. The peak intensity increased with the increase of the EMA-GMA addition, which suggested that the amorphous part of PCL in the blends became higher and higher. This was probably due to the increasing occurrence of the reaction between the EMA-GMA and PCL limiting the crystallization of PCL, which was also demonstrated in the following DSC mechanical results. 

### 3.2. Thermal and Crystallization Behaviors

Figure 5 shows the DSC second heating curves of the neat PLA, PLA/PCL and PLA/PCL/EMA-GMA blends with different amounts of EMA-GMA. PLA and PCL are both semi-crystalline polymers. The solid-state morphology and crystallinity of the blends influence their final physical properties. Consequently, it is very important and necessary to study the thermality and crystallinity of the PLA/PCL and PLA/PCL/EMA-GMA blends. The detailed data are summarized in Table 2.

As shown in Figure 5, three transitions upon heating comprise the neat PLA’s curve: a PLA glass transition, a cold crystallization peak, and a melting peak. Compared with the neat PLA, the cold crystallization peak of PLA for the PLA/PCL and PLA/PCL/EMA-GMA blends shifted toward a higher temperature. However, with the increase of the EMA-GMA content, the level of the cold crystallization peak moving toward a higher temperature decreased, and this trend was indicated by the dotted line in Figure 5. The DSC results of the PLA/PCL binary blend indicated that the dispersed PCL particles confined the crystallization of PLA of the PLA/PCL binary blend, which agreed with the finding of Thongpin et al. [34]. Based on previous discussions, EMA-GMA and PLA showed a good compatibility because of the reactive processing at a high temperature. When a small amount of EMA-GMA was added, the EMA-GMA particles further confined the crystallization behavior of PLA. But when the addition of the EMA-GMA continued to increase, the cold crystallization peak moved toward a low temperature, which means that the crystallization behavior of PLA was promoted. This phenomenon may be related to the decreased particle size of the dispersed phase, which will be further observed in the SEM results.

As shown in Figure 6, no crystallization peak of the PLA was observed in the cooling curves, for all samples, and this is attributed to the slow crystallization rate of PLA. At the same time, the crystallization peak of PCL was clearly observed in the range of −20–30 °C. For the PLA/PCL blend, a broad crystallization peak appeared near 15 °C. When EMA-GMA was added in the blend, the crystallization peak of PCL gradually shifted to a low temperature and changed from a single peak to double peaks. At this stage of crystallization, an imperfect crystalline structure was formed in some crystals. These crystals grew thicker in the heating process, corresponding to the double melting peak of the PCL in the heating curve of the PLA/PCL blend. These results indicated that the addition of EMA-GMA confines the crystallization of the PCL. As discussed above, it was confirmed that the epoxidation reaction occurred between the EMA-GMA and PCL at a high temperature. The crystallization ability of PCL decreased because the interaction between the EMA-GMA and PCL segments decreased the mobility of the PCL molecular chain. When the addition of EMA-GMA exceeded 6 phr, the width of the crystallization peak became narrow, and the area of the peak became smaller, finally stabilizing at −10 °C. These results showed that the addition of EMA-GMA confined the crystallization of the PCL but improved the perfect degree of the crystal structure. This phenomenon may be attributed to the epoxidation reaction between the EMA-GMA and PCL. According to the view of Flory and Mandelkern [35,36], the content of the end-groups of polymers had a great influence on the equilibrium melting process. A higher end-group concentration led to more dislocations during the wafer thickening process, which could lead to poor crystal perfection [37]. When the addition of EMA-GMA exceeded a certain amount, the concentration of the end-group of the PCL decreased markedly. Therefore, the crystal structure of PCL became more perfect and the double melting peaks turned into a single melting peak at a higher temperature, as shown in Figure 5.

The XRD patterns of the PLA/PCL blends with different EMA-GMA loadings are illustrated in Figure 7. The XRD patterns of the blends showed the crystalline peaks of the PCL phase overlapping with the amorphous curves of the PLA phase. The large and broad peak between 10° and 30° originated from the PLA amorphous region. Two high-intensity peaks at 2θ = 21.3° and 23.6° revealed the semi-crystalline nature of PCL. The PCL crystalline peaks’ intensity decreased significantly with the increase of the EMA-GMA addition, which was consistent with the DSC observation of a decreasing crystallinity of the PCL phase with an increasing EMA-GMA addition (Appendix A).

### 3.3. Mechanical Properties

Figure 8 shows the stress-strain curves of the selected neat PLA, binary PLA/PCL and ternary PLA/PCL/EMA-GMA blends with different EMA-GMA contents. The neat PLA was very rigid without any obvious yielding and showed a large tensile strength of about 70 MPa and a low elongation at break of only 8%, because of the natural stiffness and poor flexibility. As for the binary PLA/PCL blend, it showed a clear yielding behavior upon stretching with an elongation at break of about 80%. It was worth noting that the distinct yielding progress appeared during the tensile test and that the elongation at break drastically increased with the incorporation of EMA-GMA. The fracture behavior of the samples changed from the brittleness of neat PLA to the ductile fracture of the PLA/PCL/EMA-GMA ternary blends. In addition, another elastic and yielding deformation appearing in the developing large deformation stage of the ternary blends was analyzed in the Appendix A At the same time, as shown in Figure 9, the tensile strength and modulus of the blends decreased with the addition of the PCL and EMA-GMA, showing a similar decreasing tendency, resulting from the presence of the ductile PCL and EMA-GMA elastomer.

The notched Izod impact strength and elongation at break of the neat PLA/PCL and PLA/PCL/EMA-GMA blends are summarized in Figure 10A. As shown, the elongation at break of the PLA/PCL blend increased from 9.50% of PLA to 53.61% due to the fact that interfacial cavitation caused by debonding occurred at the interface between the components with a limited compatibility. Additionally, the interfacial cavitation would induce a multiple matrix shear yielding. However, the notched impact strength of the PLA/PCL binary blend only increased to 4.70 kJ/m^2^. Large dispersed PCL particles and a weak interaction between the interface of the PCL particles and the PLA matrix, caused by a poor compatibility, led to a limited increase in the notched Izod impact strength.

As for the ternary blends, an excellent tensile toughness was obtained for all of them, with the incorporation of EMA-GMA. As Figure 10A shows, the elongation at break increased significantly with an increase of added EMA-GMA. In particular, the PLA/PCL/8 ternary blend exhibited the highest elongation at break, at 500.94%, approximately 51.73 times better than neat PLA. The notched Izod impact strength had a similar trend as the elongation at break: the maximum notched impact strength of 64.31 kJ/m^2^ was achieved at an 8 phr EMA-GMA addition, nearly 23.26 times better than neat PLA. It should be pointed out that all of the PLA/PCL/4, PLA/PCL/6, PLA/PCL/8, PLA/PCL/10 ternary blend samples were partially fractured, as shown in the digital photograph in Figure 10B. The unfractured part of the sample would absorb more energy if it was completely fractured, so the actual value of the impact strengths of the samples would be higher than the test value, as shown by the dashed rectangle in Figure 10A. These results indicated that the PLA/PCL/8 ternary blend exhibited an optimum toughening effect compared with other formulations.

The present mechanical testing results obtained for the PLA/PCL/8 ternary blend are compared to those reported for other PLA reactive melt blending systems. In some previous similar works, Zhang et al. toughened PLA with a poly(ether-b-amide) elastomeric copolymer (PEBA) and EMA-GMA through reactive melt processing; the most toughened blend, showing an impact strength of ~500 J/m with a partial break behavior, was obtained at the optimum formulation of 70 wt % PLA, 20 wt % EMA-GMA, and 10 wt % PEBA [38]. Wu et al. also used the reactive melt blending to fabricate a supertough PLA/poly(butylene-adipate-co-terephtalate) (PBAT)/EMA-GMA multicomponent blend (75 wt %/10 wt %/15 wt %) exhibiting a significantly improved impact strength of 61.9 ± 2.7 kJ/m^2^ [39]. Xue et al. introduced EMA-GMA and poly(butylene succinate) (PBS) to toughened PLA through an in-situ reaction between different components during the melt processing; the elongation at break and impact strength reached a maximal value of 549.4% and 46.5 kJ/m^2^ when 10 wt % and 20 wt % EMA-GMA were added [40]. In addition, a special microstructure was observed for the optimum blend composition in all of the mentioned research, which suggested that the phase morphology played a key role in enhancing the toughness.

### 3.4. Morphological Analysis

The SEM micrographs of cryofracture surfaces of the PLA/PCL and PLA/PCL/EMA-GMA blends are presented in Figure 11a–f. A clear phase-separated morphology with the PCL particles dispersed in the PLA matrix was observed in Figure 11a. The clear interfacial gap between the two phases, and large voids caused by the PCL particles detaching from the PLA matrix, indicated that the PLA and PCL were immiscible. With the incorporation of 2 or 4 phr EMA-GMA, the interfaces between the PCL and PLA phases became a little obscured, although some voids caused by detaching could still be observed, which indicated that EMA-GMA could improve the compatibility of the two phases. For the PLA/PCL/6, PLA/PCL/8 and PLA/PCL/10 ternary blends, as shown in Figure 11d–f, the interface between the PCL particles and PLA matrix gradually disappeared, and the cryofracture surface became smooth with hardly any voids. Furthermore, it was difficult to distinguish the PCL particles and EMA-GMA particles from the PLA matrix as the interface between them became very indistinct. The above indicated that the compatibility of the PLA/PCL blend was greatly improved by the EMA-GMA.

To further clearly distinguish the minor phases and observe their phase structure, the cryo-fractured surfaces of the blends were etched by a solvent to eliminate the amorphousness of the PLA matrix. The SEM images of the cryofracture surfaces of the PLA/PCL, PLA/PCL/2, PLA/PCL/8 and PLA/PCL/10 blends after being etched are shown in Figure 12a–d. Additionally, Figure 12a’–d’ shows the schematic structure of the dispersed phases. As Figure 12a shows, spherical PCL particles were surrounded by PLA crystals. It could be observed distinctly that many PCL particles detached from the PLA matrix and that large voids surrounded the PCL particles, which indicated that the two phases had poor compatibility and interfacial adhesion. During the fracture progress, the poor interfacial adhesion could not prevent the crack formation, which consequently led to low mechanical properties. With the incorporation of EMA-GMA, two different morphologies were observed in the ternary blends, as shown in Figure 12b–d; one of them was the separate dispersion of the minor phases, while the other was the multiple stack formation. The EMA-GMA introduced into the blends played dual roles as both a reactive compatibilizer and a toughening agent. Consequently, some separate dispersed EMA-GMA particles can improve the toughness of the PLA as a stress concentrator, and the others can adhere to the PCL dispersed particles serving as a bridge between the PCL particles and PLA matrix and forming the multiple stack formation through the reaction occurring at a high temperature. With the increase of the EMA-GMA addition, the size of the EMA-GMA particles became smaller and smaller while the size of the PCL particles showed no significant change. The small EMA-GMA particles could promote the formation of crystal nuclei at numerous locations, which contributed to the decrease of the T_cc_ of the PLA phase corresponding to the DSC results. In addition, more and more EMA-GMA particles adhered to the PCL particles, as shown in Figure 12b’–d’. When the addition of EMA-GMA exceeded 8 phr, a large amount of EMA-GMA took part in the reaction with the PLA matrix and PCL particles; meanwhile, it was clearly observed that some dispersed stack formation connected with each other and transformed into a shish-like shape which indicated that the EMA-GMA and PCL phase had a certain degree of crosslinking, especially at a 10 phr addition of EMA-GMA. These results showed that different additive amounts of EMA-GMA had a great influence on the interfacial adhesion between the PLA matrix and PCL dispersed phase and on the morphologies of the two minor components, which will result in different toughness properties.

### 3.5. Toughening Mechanism

As is known, the efficiency of the elastomer in toughening the polymer blend significantly relies on the interfacial compatibility between different components, which determines the interfacial adhesion and the cavitation process. For elastomer toughened PLA blends, the presence of dispersed elastomer particles will initiate cavitation in the blends as a stress concentrator [41,42]. Following this, cavitation usually induces crazing or shear yielding, which further lead to a large plastic deformation of the blends as a means of consuming enormous fracture energy. There are two types of cavitation: one is internal cavitation, the other is debonding cavitation, and the interfacial adhesion between the dispersed phase and PLA matrix decides which one is dominant. A poor interfacial adhesion will lead to a phase separation, which usually causes a debonding cavitation unable to withstand the crack formation. When the interfacial adhesion is strong, internal cavitation that involves the formation of cavities in the dispersed elastomer takes place instead of debonding cavitation. However, an overly strong interfacial adhesion is also not good for toughness, as it generally delays the occurrence of matrix yielding [43]. Consequently, a suitable interfacial adhesion plays a very important role in toughening the blend.

Figure 12 shows multiple small size EMA-GMA particles stacked on the surface of PCL particles with the EMA-GMA particles between the PCL and PLA interface. The contact area at the interface between the EMA-GMA particles and the other two phases increased as the size of the EMA-GMA particles adhering to the PCL particles decreased; therefore, the interfacial adhesion became increasingly stronger along with more and smaller EMA-GMA particles adhering to the PCL dispersed particles as a bridge connecting the PLA and PCL phase. The strong interfacial adhesion limited the debonding cavitation, meanwhile propagating the internal cavitation. On the other hand, as seen from the previous analysis, the EMA-GMA particles not only connected the PLA matrix and PCL dispersed particles but also connected the PCL particles to themselves. Different PCL particles linked together to form a shish-like structure, which reduced the connection area between the PLA matrix and PCL dispersed phase; in other words, it reduced the occurrence probability of a debonding cavitation. This phenomenon became increasingly evident with the increase of the EMA-GMA addition, as shown in Figure 12. 

The SEM images of the impact fracture surfaces of the PLA/PCL and PLA/PCL/EMA-GMA blends are presented in Figure 13a–f. In Figure 13a, the impact fracture surface of the PLA/PCL blend was smooth without much deformation and showed many spherical voids caused by the debonding of the dispersed PCL particles, which indicated a typical brittle fracture resulting from the poor interfacial adhesion between the PCL particles and PLA matrix. With the incorporation of 2 phr EMA-GMA, there were some discernible fibrils on the fracture surfaces, while the voids resulting from the pullout of the PCL particles still existed, as shown in Figure 13b, indicating an improved toughness of the blend. When 4 phr EMA-GMA was added to the PLA/PCL blend, a large amount of plastic deformation caused by the debonding cavitation of the PCL dispersed particles was observed (as shown in Figure 13c) because of the further improved interfacial adhesion. With the addition of 6 phr EMA-GMA, the debonding cavitation was limited and replaced by some internal cavitation which formed some microfibrillar structures of the EMA-GMA phase. When the addition of EMA-GMA exceeded 8 phr, as Figure 13e,f shows, the impact fracture surfaces of the blends exhibited numerous cavitation and entangled fibril structures along the crack propagation, showing a significant toughening effect. Not only the debonding cavitation but also the internal cavitation of EMA-GMA, which could have formed cavities with internal microfibrillar structures, appeared in the impact fracture surface of the PLA/PCL/8 formulation. Compared with the PLA/PCL/10 blends, there were more cavities with different sizes distributed in the PLA/PCL/8 blend.

As discussed previously, the PLA/PCL/8 formulation exhibited a superior toughness compared to the other formulations. A combination of the debonding cavitation and internal cavitation, which will easily induce the plastic deformation of the surrounding matrix, was observed in the impact fracture surface of PLA/PCL/8, as shown in Figure 13e. This phenomenon helps to efficiently dissipate the fracture energy with plenty of cavities with varying sizes emerging during the development of the fracture progress. For the PLA/PCL/8 ternary blend, the delicate balance of the debonding cavitation and internal cavitation under an optimum interfacial adhesion and a suitable phase morphology was believed to play an extremely favorable role in toughening the PLA matrix. 

## 4. Conclusions

In this work, supertough biodegradable PLA/PCL blends with the addition of EMA-GMA were successfully prepared by reactive melt blending. An FT-IR analysis showed that the epoxy groups of the EMA-GMA reacted with the carboxyl and hydroxyl end groups of PLA and PCL. Melt torque and DMA analyses proved the interfacial compatibilization effects of EMA-GMA on PLA ternary blends. DSC and XRD analyses revealed that the introduction of EMA-GMA will influence the crystallinity of PLA and reduce that of the PCL phase due to reactive interfacial compatibilization. We obtained a supertough PLA/PCL/EMA-GMA blend, showing a notched impact strength of 64.31 kJ/m^2^ and an elongation at break of 500.94%, at an 8 phr addition of EMA-GMA. Different interfacial adhesions and phase morphologies of the ternary blends, which determined the toughening effect, were controlled by the addition of EMA-GMA. A suitable multiple stacked phase structure with an optimum interfacial adhesion was achieved for the PLA/PCL/8 ternary blend. Morphological studies indicated that the synergistic effect of the debonding cavitation and internal cavitation, followed by the large-scale plastic deformation of the PLA matrix, was responsible for the super toughening effect.

## Figures and Tables

**Figure 1 polymers-11-00771-f001:**
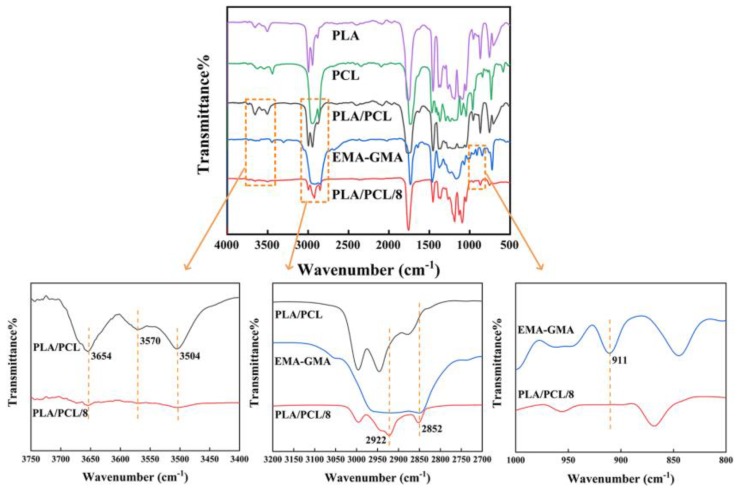
FT-IR spectra of the PLA, PCL, EMA-GMA, PLA/PCL and PLA/PCL/8 blends.

**Figure 2 polymers-11-00771-f002:**
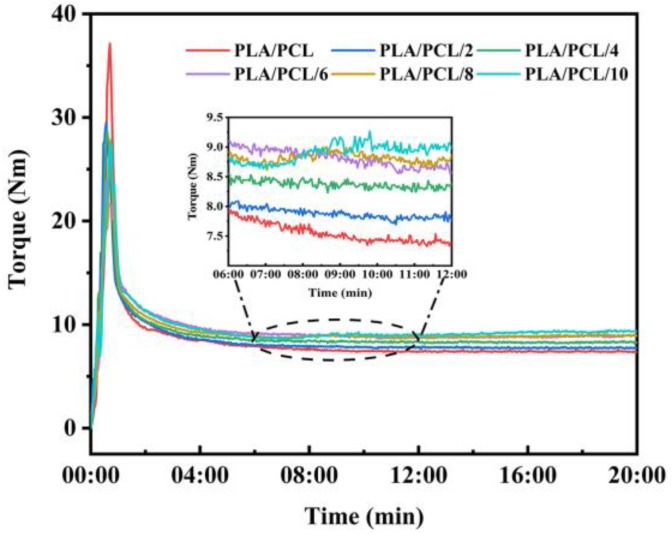
Melt torque during the reactive blending of the PLA/PCL blends with different EMA-GMA additions.

**Figure 3 polymers-11-00771-f003:**
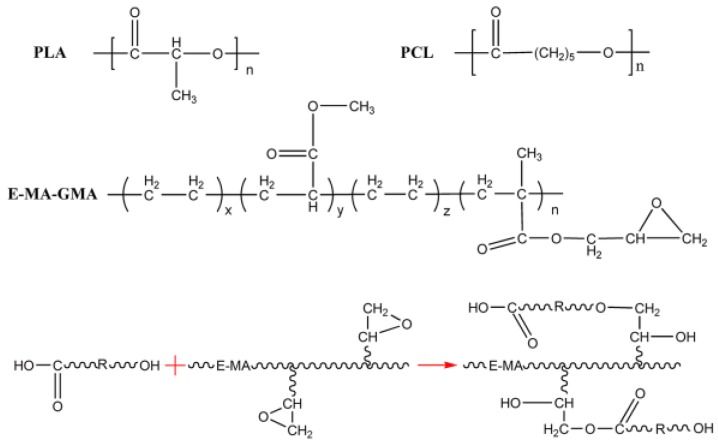
Schematic illustration of the possible reactions in the PLA/PCL/EMA-GMA blend (“R” represents PLA or PCL).

**Figure 4 polymers-11-00771-f004:**
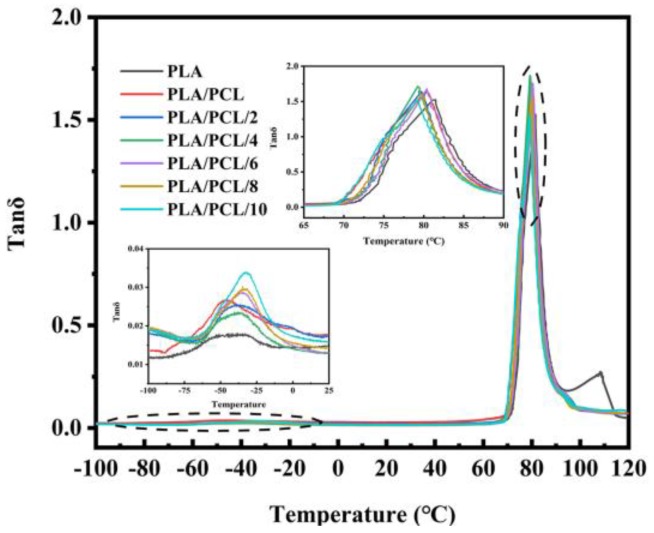
Plots of the loss factor (tan δ) against the temperature for the neat PLA, PLA/PCL and PLA/PCL/EMA-GMA blends.

**Figure 5 polymers-11-00771-f005:**
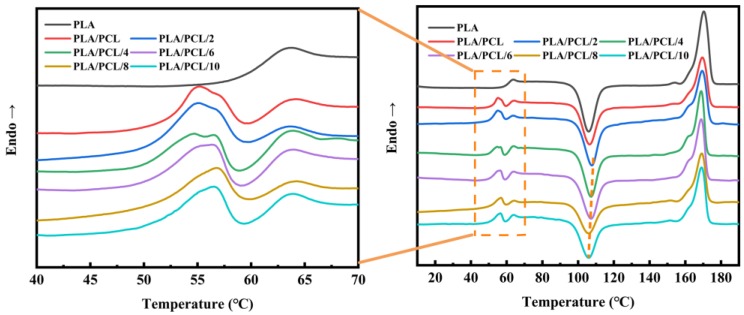
DSC second heating curves of the neat PLA, PLA/PCL and PLA/PCL/EMA-GMA blends.

**Figure 6 polymers-11-00771-f006:**
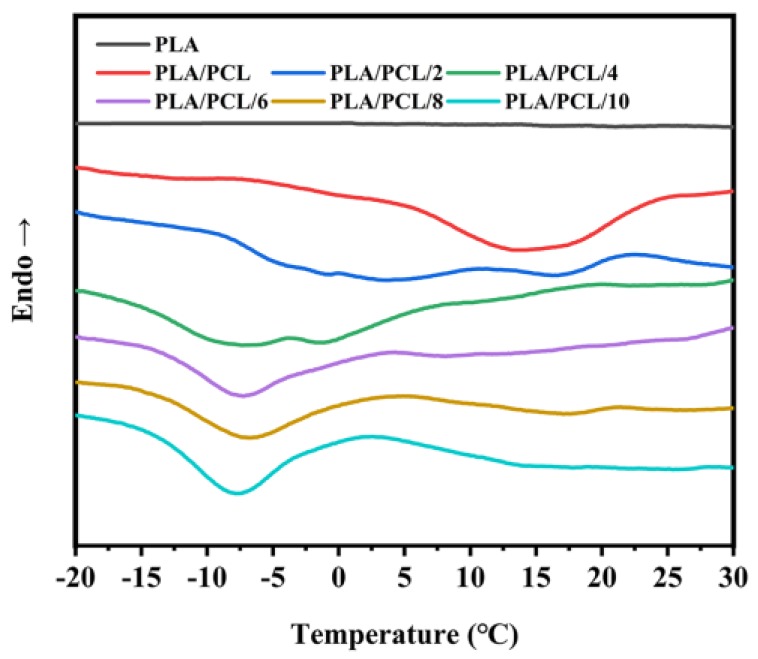
DSC cooling curves of the neat PLA, PLA/PCL and PLA/PCL/EMA-GMA blends in the range of −20–30 °C.

**Figure 7 polymers-11-00771-f007:**
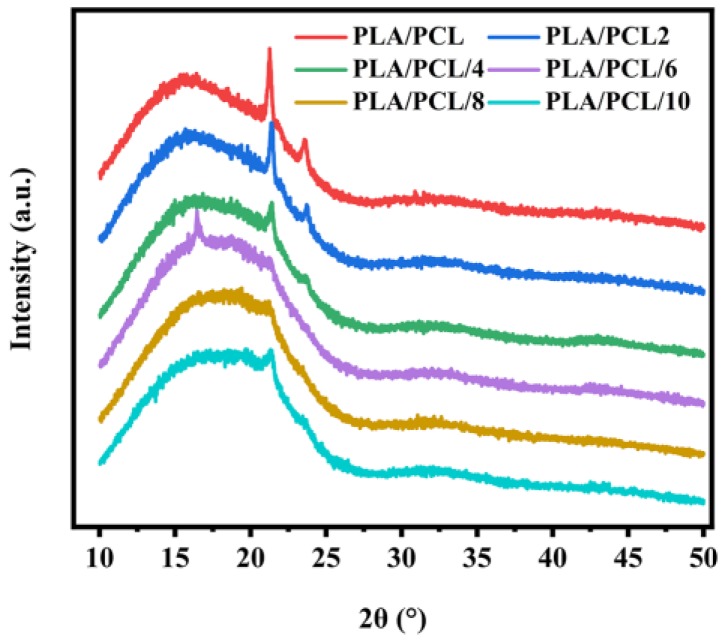
XRD patterns of PLA/PCL with different additions of EMA-GMA.

**Figure 8 polymers-11-00771-f008:**
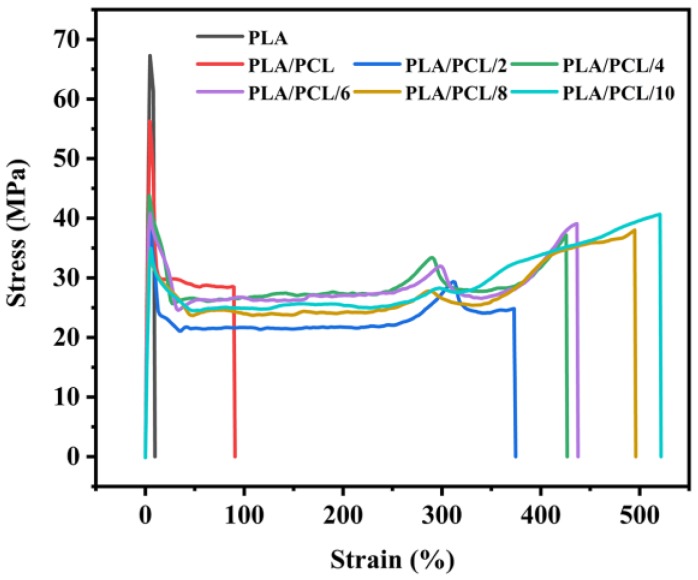
Tensile stress-strain curves of the neat PLA, PLA/PCL and PLA/PCL/EMA-GMA blends.

**Figure 9 polymers-11-00771-f009:**
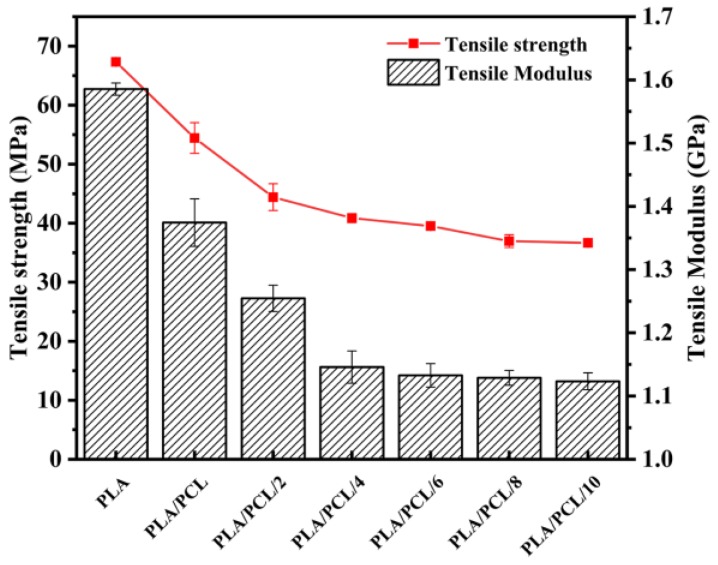
Tensile strength and tensile modulus of the neat PLA, PLA/PCL and PLA/PCL/EMA-GMA blends.

**Figure 10 polymers-11-00771-f010:**
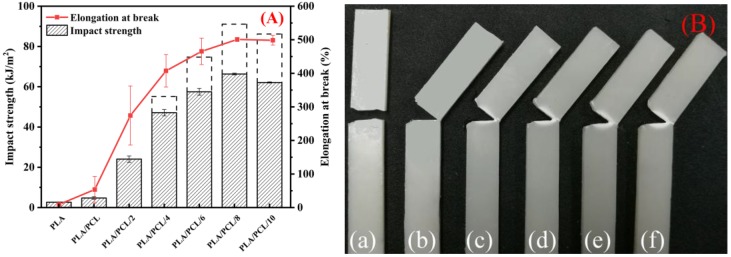
(**A**) Notched Izod impact strength of the neat PLA, PLA/PCL and PLA/PCL/EMA-GMA blends; and (**B**) digital pictures of the fractured samples: (a) PLA/PCL; (b) PLA/PCL/2; (c) PLA/PCL/4; (d) PLA/PCL/6; (e) PLA/PCL/8; and (f) PLA/PCL/10.

**Figure 11 polymers-11-00771-f011:**
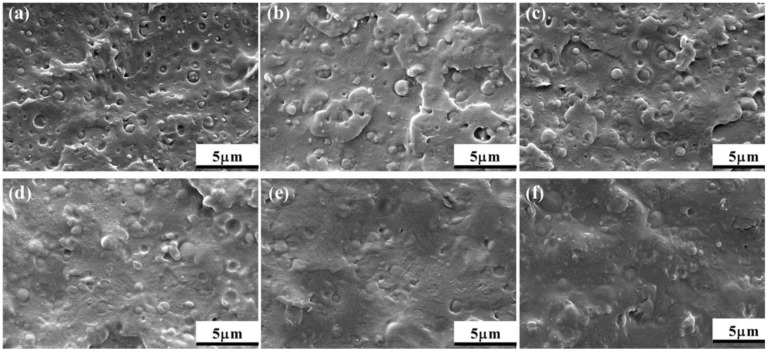
SEM micrographs of the cryofracture surfaces: (**a**) PLA/PCL; (**b**) PLA/PCL/2; (**c**) PLA/PCL/4; (**d**) PLA/PCL/6; (**e**) PLA/PCL/8; and (**f**) PLA/PCL/10.

**Figure 12 polymers-11-00771-f012:**
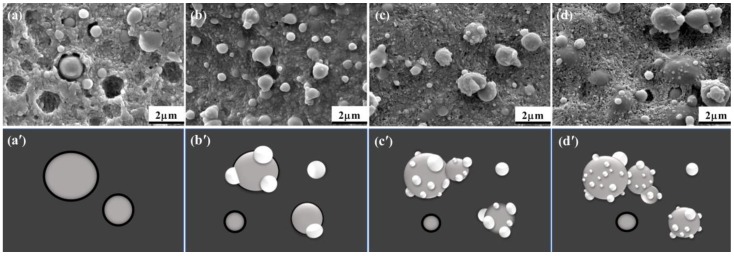
SEM images of the cryofracture surface after being etched and the schematic phase structure of (**a**,**a’**) PLA/PCL; (**b**,**b’**) PLA/PCL/2; (**c**,**c’**) PLA/PCL/8; and (**d**,**d’**) PLA/PCL/10.

**Figure 13 polymers-11-00771-f013:**
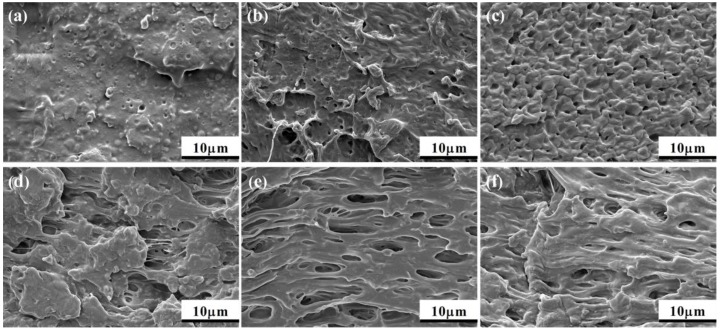
SEM images of the impact fracture surfaces: (**a**) PLA/PCL; (**b**) PLA/PCL/2; (**c**) PLA/PCL/4; (**d**) PLA/PCL/6; (**e**) PLA/PCL/8; and (**f**) PLA/PCL/10.

**Table 1 polymers-11-00771-t001:** DMA results of the PLA, PLA/PCL and PLA/PCL/EMA-GMA blends.

Samples	*T*_g1_ (°C)	*T*_g2_ (°C)
PLA	-	81.4
PLA/PCL	−46.8	80.8
PLA/PCL/2	−38.6	79.7
PLA/PCL/4	−37.0	79.3
PLA/PCL/6	−34.5	80.4
PLA/PCL/8	−32.5	79.7
PLA/PCL/10	−32.0	79.0

**Table 2 polymers-11-00771-t002:** Thermal properties of the neat PLA, PLA/PCL and PLA/PCL/EMA-GMA blends.

Samples	*PLA*						*PCL*		
	*T*_g_ (°C)	*T*_cc_ (°C)	ΔH_cc_ (J/g)	*T*_m_ (°C)	ΔH_m_ (J/g)		*T*_c_ (°C)	*T*_m_ (°C)	ΔH_m_ (J/g)
PLA	63.8	105.5	27.45	170.3	38.49			-	
PLA/PCL	64.2	106.5	22.46	169.5	26.73		13.5	55.2/56.8	2.819
PLA/PCL/2	63.7	107.9	23.94	169.5	27.51		3.6/16.6	55.1/56.5	2.914
PLA/PCL/4	63.8	107.5	24.71	168.9	28.4		−7.3/−1.2	54.7/56.5	2.834
PLA/PCL/6	63.9	107.3	25.43	168.9	30.05		−7.2	56.3	2.639
PLA/PCL/8	64.2	106.1	19.11	169.2	25.17		−6.9	56.8	2.282
PLA/PCL/10	63.8	105.8	21.78	169.1	28.34		−7.8	56.5	2.315

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
