# Peer review of "Super-Toughened Poly(lactic Acid) with Poly(ε-caprolactone) and Ethylene-Methyl Acrylate-Glycidyl Methacrylate by Reactive Melt Blending"

_polymers, 2019, doi:10.3390/polym11050771_

Round 1

Reviewer 1 Report

This article is devoted to the synthesis and investigation of thermal and mechanical properties of Poly(lactic Acid) with Poly(ε- caprolactone) and Ethylene-Methyl Acrylate-Glycidyl Methacrylate polymer. The polymer was synthesized by melt blending method. In particular, FTIR method was applied to study chemical reaction between PLA, PCL and EMA-GMA. Differential Scanning Calorimetry and Dynamic Mechanical Analysis were used to study miscibility and thermal properties of the system. X-ray Diffraction and Scanning Electron Microscope were used to investigate the phase composition and morphology of the samples. The authors aimed to improve toughness of initial biodegradable PLA polymer. 

I would like to recommend the article for publication. I have also one comment:

Figure 1 shows FTIR spectra of the initial and prepared polymers. There are many various peaks in those spectra. Authors only explain some of them. It would be interesting to the readers to know how other peaks may be assigned taking into account that its positions and relative intensities depend on polymer composition. 

Author Response

Dear reviewer:

Thank you very much for your comment for the manuscript dated 07 April. According with your comment, we amended the relevant part in manuscript. Your comment was replied below:

Point 1:

Figure 1 shows FTIR spectra of the initial and prepared polymers. There are many various peaks in those spectra. Authors only explain some of them. It would be interesting to the readers to know how other peaks may be assigned taking into account that its positions and relative intensities depend on polymer composition.

Response 1:

Thank you for your suggestion, just like what you said, we only explained the characteristic peak related to the reaction in the original manuscript. In order to make the FT-IR analysis more comprehensive, the other peaks also need to be explained. We have supplemented the explanation of the other peaks in FT-IR analysis. (Line 147-152, Page 4).

Thank you for your kind suggestion!

Yours sincerely,

Ao-Lin Hou, Jin-Ping Qu

School of Mechanical and Automotive Engineering,

South China University of Technology,

Guangzhou, 510640, Guangdong, China

Reviewer 2 Report

This work aimed at ternary blend of poly(lactic acid) (PLA) with poly(ε-caprolactone) (PCL) and ethylene-methyl acrylate-glycidyl methacrylate (EMA-GMA). The basic idea of the paper sounds interesting, but the phase behavior of PLA/PCL/EMA-GMA blends was anticipative results. In addition, there are some unclear results in the further discussions between ternary blend and intermolecular interactions. Thus, I do not think that it is acceptable in its present form. I would like to recommend the authors discuss more detailed and revise this paper.

1. In blending analysis, the reviewer cannot see any explanation of the effect of molecular weights on the melt-blended polymer composites. Molecular weight is an important factor that can influence the performance of polymer blends.

2. In addition, 13C NMR spin–lattice relaxation times T1 and small/wide angle X-ray scattering instruments are quite important for phase transition and microstructural features of polymer blends. These results should be included in the main text

3. The resulting stress-strain curves should be included in the Supporting Information.

4. Hardness analyzer is also suggested to be used to identify differences in impact resistance between PLA/PCL and PLA/PCL/EMA-GMA blends.

5. In Figure 2, it is more persuadable to include detail discussion of polymer-polymer interactions on the miscibility and the phase structure of PLA /PCL/EMA-GMA blends.

6. Finally, authors do not compare their processing route and mechanical performances with preexisting literature results.

Author Response

Dear reviewer:

Thank you very much for your comments dated 07 April. Your suggestions have enabled us to improve our work. According with your advice, we have made extensive modification on the original manuscript. Here, we attached a document replying your comments and revised manuscript for your approval.

Once again, special thanks to you for your good comments!

Yours sincerely,

Ao-Lin Hou, Jin-Ping Qu

School of Mechanical and Automotive Engineering,

South China University of Technology,

Guangzhou, 510640, Guangdong, China

Round 2

Reviewer 2 Report

After carefully reading the authors' response to my original comments and revised manuscript, I recommend this paper for publication without further revision.